# Passive Voice Comprehension during Thematic-Role Assignment in Russian-Speaking Children Aged 4–6 Is Reflected in the Sensitivity of ERP to Noun Inflections

**DOI:** 10.3390/brainsci12060693

**Published:** 2022-05-27

**Authors:** Olga Kruchinina, Ekaterina Stankova, Diana Guillemard, Elizaveta Galperina

**Affiliations:** Sechenov Institute of Evolutionary Physiology and Biochemistry of the Russian Academy of Sciences, 194223 Saint-Petersburg, Russia; kruchinina_ol@mail.ru (O.K.); stankova.ekaterina@yandex.ru (E.S.); diana-tsap@yandex.com (D.G.)

**Keywords:** event-related potentials (ERP), thematic-role assignment, children, passive voice acquisition, frontocentral positivity, LAN, P600, NP2

## Abstract

Children tend to rely on semantics rather than syntax during sentence comprehension. In transitive sentences, with no reliance on semantics, the syntax-based strategy becomes critical. We aimed to describe developmental changes of brain mechanisms for syntax processing in typically developing (TD) four to six year old’s. A specially designed sentence-picture matching task using active (AV) and passive (PV) voice enforced children to use grammar cues for sentence comprehension. Fifty children with above >60% level of accuracy in PV sentences comprehension demonstrated brain sensitivity to voice grammar markers-inflections of the second noun phrase (NP2), which was expressed in a greater event-related potentials (ERP) amplitude to PV vs. AV sentences in four-, five-, and six-year-old children. The biphasic positive-negative component at 200–400 ms was registered in the frontocentral and bilateral temporoparietal areas. Only in six-year-old children P600 was registered in the right temporoparietal area. LAN-like negativity seems to be a mechanism for distinguishing AV from PV in the early stages of mastering syntax processing of transitive sentences in four to five year old children. Both behavioral and ERP results distinguished six-year-olds from four-year-old’s and five-year-old’s, reflecting the possible transition to the “adult-like” syntax-based thematic role assignment.

## 1. Introduction

### 1.1. Syntax in Speech Comprehension

“Who did what to whom”—is the key question for successful sentence comprehension. The development of a child’s language mainly follows the path of improving the processes of establishing thematic relationships of words in a sentence. Speech comprehension is based on context and semantics and is verified by syntax [1,2,3]. Therefore, the mechanisms of semantic analysis are formed earlier than those of syntactic analysis in ontogenesis [2,4,5,6]. When it is impossible to unambiguously interpret thematic roles relying on semantics (for example, in transitive sentences with both animated subject and object), reliance is made only on grammatical cues such as word order, voice, case marking, etc. [6,7]. Sentence comprehension is realized differently between the ages of two and four, when word order is critical for children, and between the ages of five and seven, when children rely more on upon case marking for sentence interpretation [6,7,8]. Children can detect grammatical case-marking cues, by the age of three, but they start using this information to determine who is doing what to whom in a sentence only at the age of six [9,10,11]. In a situation of conflict between two cues, word order and case marking, children could rely on case marking over word order only by the age of seven [7]. Understanding complex syntax can be difficult for children as young as six, nine or even 12 years of age [12,13,14,15]. Semantic-independent syntax activity is only established at ages 9 to 10 [16]. Thus, the improvement of syntax skills continues for a long time in ontogeny and may depend on the language.

### 1.2. Language-Specific Syntax

Thematic role encoding varied across different languages: with word position in strong word order languages, like English, or case marking in free word order languages like German or Russian [17,18]. In the Tagalog language, the verb is inflected for voice, aspect, and mood, while the word order is the same for passive and active sentences, so the roles of agent and patient is defined by changing the voice marking of the verb [19]. Special passive markers like BEI in Mandarin help to interpret correctly thematic roles. Thus, Huang and colleagues [20] showed that five-year-old’s correctly interpreted passives more often when the passive marker BEI (indicates that the first noun phrase is a patient) appeared after a pronoun, compared to when the marker appeared after a referential noun. In German, simple transitive structures, the definition of a noun as a subject or an object depends on its case-marking (Der/Den) at the beginning of the nominative phrase (NP1 and NP2) [10].

In Slavic languages like Russian, Ukrainian, and others, word order flexibility is provided by the rich inflectional case morphology that establishes the relationship between the different constituents in a sentence [21,22,23]. In Russian, the same semantic situation can be expressed with active (AV, Example (a)) or passive (PV, example (b)) voice using appropriate case marking:

NP1VNP2
(noun phrase 1)(verb)(noun phrase 2)(a)Active voice(AV)Subject (noun)Action (verb)Object (noun)“The grandson (Nom)huggedthe grandfather (Acc).”/Vnukobnja***l***Ded***a***./(b)Passive voice(PV) Object (noun)Action (participle)Subject (noun)“The grandson (Nom)is hugged bythe grandfather (Inst).”/Vnukobnja***t***Ded***om***./

Therefore, in Russian, to assign thematic roles in transitive sentences, it is necessary to analyze grammatical markers—noun inflections and participle suffixes in cases when it is impossible to rely on semantics. There is no direct analogy in other languages for expressing the passive voice in Russian; however, it is possible to compare some results in case markings, taking into account their position—prefixes or infixes, while in Russian it is inflections. Despite the linguistic differences in the expressing of thematic roles in different languages, the brain mechanisms for their establishment may have similarities, as well as specific nuances that reflect the characteristics of a particular language. 

The mastering of grammar continues throughout preschool and primary school age in TD children and depends on the frequency of grammatical tools in the language [24,25,26]. The case sensitivity in an adult-like manner was shown in Russian children as young as 4;2 (four years and two months) [22], while the acquisition of passive is sometimes not finished as late as the age of seven [27]. The frequency of the grammar tool in the child’s input could be critical to its age of acquisition (AoA) [28], so AoA varies greatly for the passive through languages [19,25,29,30,31].

Canonical word order sentences, simple inflectional paradigms, and detection of case marking violations are already mastered by three-year-old children [9,32,33], but they still have difficulty distinguishing object-subject relations [10]. The use of case marking for thematic-role assignments has been shown in children from the age of four or five [6,7,10]. Complex syntactic structures like passive- and object-first sentences are only acquired between four and seven years of age [7,11] but even seven- or eight-year-old children can make mistakes, especially in complex, transitive sentences with passive voice or reversed word order [27,34,35]. 

The comparison of neurophysiological results requires keeping in mind the language specifics because different grammatical markers in different languages could serve as a trigger for the brain’s response. 

### 1.3. Neural Correlates to the Complex Syntax Processing

The most frequent areas mentioned as activated in the analysis of semantics and syntax during sentence processing are the frontal and temporoparietooccipital areas on the left, as well as the temporal cortex area on the right [10,18,36,37,38]. The above areas include both nonspecifically activated areas during processing different types of verbal information and those that are activated specifically during processing semantics and syntax. The posterior part of the left superior temporal gyrus and sulcus, as well as the posterior part of the left medial temporal gyrus, are specifically activated during the analysis of syntax [33]. Analysis of the verb-argument relationship is provided by the interaction of the above structures with the pars opercularis of the left inferior frontal gyrus [17,39,40]. The connection of the posterior temporal areas of the brain with the ventrolateral cortex is provided by the dorsal stream pathway of speech processing, in particular, the upper longitudinal and arcuate fascicles [16,40]. In addition, the anterior portion of the left superior temporal gyrus (left aSTG) has been discussed as being involved in syntax processing of sentences [41,42], as well as in combinatorial semantics [43]. Making thematic judgments about sentences recruits the inferior frontal gyrus extending to the dorsolateral prefrontal cortex, regions that remained inactive during passive comprehension of the same sentences [44]. These findings highlight the flexibility of sentence comprehension neural processes and demonstrate that activation patterns can rapidly change to meet changing task demands.

Brain mechanisms for sentence comprehension encoding subject-object relations have been described mainly in adults [45,46,47]. The exploration of the developmental dynamics of higher-order sentence-level mechanisms is still in its beginnings [40]. Two main components are used in relation to morphosyntactic violations: LAN (sometimes N400) and P600 [48,49]. 

Early left anterior negativity or ELAN is observed during automatic reconstruction of phrases in adults [50]. In children it is registered from 12–13 years of age [51,52]. It is likely that younger children process morphological and syntactical errors as semantic disturbances, since they show widespread negativity between 100 and 300 ms, i.e., something between ELAN and N400 [51,53]. As a rule, the N400 component in children has a higher amplitude, greater latency, and is more widespread than N400 in adults [54,55].

In three-word transitive constructions (NP1-V-NP2) the ERPs revealed a negativity peaked around 300 ms for the processing of the topicalized accusative marked noun phrase (NP1) [10].

Three-year-old German children rely on semantics and show no brain sensitivity to markers of subject-object relations; 4.5-year-old children rely on the second noun (NP2) in the sentence thematic role assignment, showing a frontocentral ERP positivity effect indicating difficulties in syntactic integration, while six-year-old children already rely on the first noun (NP1), showing the negativity effect typical of adults [4,10]. Another brain effect—a positivity (300–500 ms) time-locked to NP1 for object-initial compared to subject-initial structures and a biphasic early positivity (220–600 ms) and late negativity (750–1200 ms) time-locked to NP2 was described in German three-year-old’s [56]. The processing of the NP1 in the object-first sentences compared to subject-first sentences was associated with an early positivity in two- to three-year-old children [4].

In German, children aged 48 months (four years old) early frontal positivity differences peaking at about 300 ms were shown during comprehension of normal/anomalous words [57]. The authors considered that frontal positivity effect in children overlapped with the LAN-effect, expected in adults, which could represent the same cognitive process as LAN, but given a different developmental stage of the cytoarchitecture of Broca’s area [58,59], was generated by different brain electrical activity. In adults, LAN was reported at both NP1 and NP2 in response to non-canonical unambiguous sentences [60].

It was shown that at the age of three to six years there is an intensive development of neural language networks that occurs: the thinning of the left frontal cortex [40], strengthening of structural and functional connections of the inferior frontal and temporal areas of the left hemisphere [40,59]. Children of three and six years of age with the above chance level of accuracy in thematic roles assignment in German demonstrated age differences in FC between left pSTG and the left IFG [59]. Thus, similar behavioral outcomes in children three and six years old are provided by different brain mechanisms. 

At the age of three to four years, the temporal areas of the left hemisphere, which are activated in the analysis of semantics and syntax, overlap significantly in children [40]. From the age of six, syntactic analysis recruited additional special frontal and temporal areas. 

### 1.4. The Current Research

The developmental trajectory of passive voice comprehension in preschool children is still not described in detail. 

The aim of our project is determined by four points: (i) children tend to rely on semantics, not grammar, during sentence comprehension until they are nine years old [16]; (ii) at the age of three to six years, there is intensive maturation of the neural networks providing syntactic processes [40]; (iii) skill improvement leads to an intensive restructuring of the neural network and affects the ERP [61]; and (iv) passive voice comprehension in Russian children is the most intensive at the age of four to six [27]. The questions are: is it possible to find specific markers of the passive voice processing? how is the intensive improvement of the syntax-based strategy of passive voice comprehension in children reflected in ERP’s? how will these markers change with an increase in age and skills improvement? 

To answer all these questions, we need to study brain correlates of passive voice comprehension in adults and children in a wide range of ages (4–10 years) with different comprehension levels, including the longitudinal study. In order to focus on the syntax, we have designed stimuli that critically reduced the possibility of relying on semantics in comprehension. We established paired mirror pictures with transitive actions. Both subject and object of the action were well-known characters, of the same gender, of approximate size, etc. Also, stimuli were balanced in parameters that are known to affect ERP’s: word frequency and length, number of syllables, audio duration, colors and size of pictures [9]. Therefore, the design of our experiment allows us to assess the reliance on syntax without a reliance on semantics.

As a first step, we studied adults and children of four to five years of age that were distinguished by their comprehension of the passive voice: high (>80%) and low (40–60%) levels of comprehension [62]. In both adults and children with a high level of correct thematic role assignment in PV (>80%), cerebral sensitivity to syntactic markers of grammar voice (distinctive morphemes: inflections or suffixes) was shown. Cerebral sensitivity was expressed in AV/PV ERP amplitude differences during the NP2 processing (positivity at 200–600 ms, with the beginning in frontocentral areas spreading to the temporoparietal areas bilaterally). In children with a low passive voice comprehension (40–60%) there were no ERP differences in the temporoparietal regions of the left hemisphere in contrast to adults and peers with a high level of PV comprehension. Thus, the functioning of the language areas of the left hemisphere is critical to ensure understanding of complex syntactic constructions, such as the passive voice. Since both behavioral and ERP results of four to five-year-olds varies greatly, we decided to split the group and assess age changes in a narrow age window—one year. 

In this study, we aim to describe in detail the developmental changes of cerebral mechanisms for the syntax-based thematic role assignment in children four to six years old who have already mastered the skill of PV comprehension (correct answers to PV sentences above 60%); we formed separate groups of children that were four, five and six years old. We assumed that children in these groups will show the PV grammar markers sensitivity—ERP differences in the frontal and temporoparietal regions of the left hemisphere, and as the skill matures and improves, the switch to the adult-like activation with greater involvement of frontal areas will occur.

We hypothesized that functional sensitivity to the passive voice could be obtained as a difference of ERP amplitude on inflections of the verb (V) and last noun (NP2) in a three-word sentence in “language” areas of the left hemisphere. We expect the ERP effect on 200–400 ms in frontal and temporoparietal areas, especially in the left hemisphere. It could be a positive or biphasic component in LAN-like latency. Subtle changes in ERP latency and amplitude difference between AV and PV will become apparent with age.

## 2. Materials and Methods

### 2.1. Subjects

Preschool children four to six years of age (*n* = 71, 41 female) were recruited from St. Petersburg kindergartens. All participants were typically developed (TD), monolingual Russian children without any hearing or neurological problems. Information on the child’s handedness was obtained by a questionnaire in advance of the study. All children participated in a language development test [40] to exclude developmental language disorder (DLD). A combined psychological and speech examination was conducted to confirm TD in children. All children included in our behavioral and ERP samples had normal speech development according to the standardized values [41]. Written informed parental consent was obtained for all children. All children participated in the study voluntarily. At the end of the study, all children received gifts. All studies were conducted following the principles of biomedical ethics outlined in the 1964 Declaration of Helsinki and its subsequent updates and was approved by the Commission on the Ethics of Biomedical Research of the Sechenov Institute of Evolutionary Physiology and Biochemistry RAS (protocol 1-07 of 16 July 2019).

### 2.2. Speech Examination

In the Russian speech therapy school, a child’s speech development is assessed as TD or developmental language disorder (DLD) [63]. We used the scaled Zajceva, Sheptunova test [64] for a ranged assessment of different speech components (phonological awareness and lexical competence). The phonological awareness was assessed by repeating syllables and words with similar phonological and articulatory patterns (12 series of syllables, 12 pairs of words), finding a given phoneme from the word, naming the first and the last phoneme in the word (five tasks each). The lexical competence was assessed by evaluation of expressive and impressive speech development: the child’s vocabulary (naming words from different semantic groups) and syntactic awareness (word-building and accidence, six types of tasks, five in each). We considered the performance of all the tasks in each section as 100% and calculated the percentage of correctly completed tasks, which was interpreted as the degree of formation of the corresponding component of the speech system.

Psychological testing included a nonverbal intelligence test (Raven’s Color Progressive Matrices) [65], as well as assessing the working memory and attention using subtest 5 of the Wechsler questionnaire (forward Digit span) [66]. 

### 2.3. Materials

The brain response during the performance of a picture-sentence matching task varies depending on the skill level of thematic-role assignment in PV sentences. To describe it correctly, we had to develop the stimulus material, which on the one hand, could specifically assess a child’s usage of grammatical markers, and on the other, the brain sensitivity to ERP components, reflecting this skill. Therefore, when developing the stimulating material, we tried to counterbalance both the lexical material and the physical characteristics of stimuli (audio files and images) according to the parameters that affect the characteristics of the ERP [47].

#### 2.3.1. Linguistic Stimuli

Pairs of animate nouns (*n* = 50) were selected as the object and subject of the action, which were balanced by gender (the same gender in sentences), number (only single), number of syllables (two to three) and frequency factor. The frequency of nouns, verbs and participles no less than 10 items per million (ipm), according to the corpus of the Russian language [67]. Four types of sentences (NP1-V-NP2, total amount 268) were designed: active voice, direct word order (for example, “The grandson hugged the grandfather—Vnuk (Nom) obnjal deda (Acc); passive voice, direct word order (“The grandson is hugged by the grandfather—Vnuk (Nom) obnjat dedom (Inst); active voice, reversed word order (“The grandson hugged the grandfather—Deda (Acc) obnjal vnuk (Nom)”) and passive voice, reversed word order (“The grandson is hugged by the grandfather—Dedom (Inst) obnjat vnuk (Nom)”). Only 2 types of direct word order sentences were analyzed in this study—AV and PV. 

#### 2.3.2. Visual Stimuli

The pictures describing situations were created using Adobe illustrator and showed the two characters (humans or animals) participating in an action. Each character occurred equally often as subject-object (agent and patient), an example image is shown in Figure 1. The following principles guided the development of the images: (1) The subject and object of the action are clear and distinguishable, as is the action itself; (2) There are no unnecessary details in the image that are not relevant to the story; (3) Paired images are color-balanced; (4). Subject and object occupied a comparable image area (a difference of no more than 25%), in order to avoid treating the larger character as the subject; (5) The number of colors used was no more than four on each pair of pictures; and (6) The direction of the action is symmetrical on each pair of pictures. The procedure of validation was the same as for the sentences. 

#### 2.3.3. Sound Files

All sentences were recorded in a 62.5 m^3^ (5 × 5 × 2.5 m) sound-insulated anechoic chamber by a female speaker with a clear articulation. The female voice was chosen as the most frequently heard by younger children. Also, changes in the pitch and tone of the voice could affect the ERPs. The external noise attenuation was at least 40 dB in the frequency range 0.5–16 kHz. A Rhode NT-USB condenser microphone with a cardioid directional pattern and built-in ADC (sampling frequency 44,100 Hz, 16 bits) was used for recording. The microphone was connected via a USB port to a PC ASUS Sonic Master with the sound editor Adobe Audition 1.5, on which we recorded and edited audio files. Sound files were saved in WAV format (44,100 Hz, 16 Bit). After recording, sentences were digitized (44.1 kHz/16 bit sampling rate, mono) and normalized in amplitude to 70%. Afterward, word durations are aligned separately for the first (NP1), second (V), and third (NP2) words in a sentence (these parameters are the most critical for ERP research). The duration of all sentences accounted for 3100 ms (see Figure 1) and did not differ between conditions.

#### 2.3.4. Validation 

In order to exclude images and audio stimuli that could be ambiguously interpreted by the subjects from the study, the stimuli were validated. We tested images (subjects and stories) and sentences on separate groups of children and adults. The naming test pilot study has been performed on adults (*n* = 20, 23–25-year-olds, six males) and children (*n* = 20, four to six-year-olds, 10 males) separately in order to determine the names of images (subjects), to calculate the scores for name agreement and image agreement. The items that had an error rate greater than 50% in the adult group were removed from the list of stimuli. The images with the mean percentage of name agreement score of about 80% in the children’s group were selected as stimuli for this study. In a special experimental series with both adults (*n* = 38, 23–27-year-olds, 16 males), and children (*n* = 62, four-to-seven-year-olds, 28 males) the experimental paradigm used in this study—paired story pictures with audio files and an adequate procedure including presentation and response timing—was tested. Due to COVID 19 lockdown, the stimulus material was tested in an online format on the https://pavlovia.org/platform (accessed on 22 March 2022). For the further study, we selected 268 stimuli (sentences and corresponding images) that satisfy the following conditions:No more than 10% of children made mistakes in comprehending the situation expressed in the active voice.No more than 10% of adults made mistakes in the same situations expressed in the passive voice.Only those situations that corresponded to both of the first conditions were chosen.Only those situations that corresponded with all three conditions also in the reversed situation: four sentences for the situation “The grandfather hugged the grandson: and four sentences for the “The grandson hugged the grandfather”.

### 2.4. Procedure

The EEG was continuously recorded from silver-silver chloride electrodes at 31 sites (according to the 10–20 International System of Electrode Placement) in the band 0.53–70 Hz, sampling frequency 250 Hz per channel, using the united ear electrode as a reference, the ground electrode was placed on the subject’s head between the electrodes *FCz* and *Fz* (Mitsar-EEG-ERP31/8, St-Petersburg, Russia). The electrodes were secured in an elastic electrode cap. An EOG was recorded from two electrodes at the outer canthi of both eyes (horizontal EOG) and from single electrodes on the infraorbital and supraorbital ridges of the right eye (vertical EOG). Electrode impedances were kept below 5–10 kΩ. EEG data were digitized online at a rate of 250 Hz and stored on a hard drive for further analysis.

During the EEG experiment, children sat in a comfortable chair in an electrically shielded and sound-attenuated EEG booth. The parents were seated next to the child, but they could not see the images or hear the sentences, and the child could not see their faces. Parents were instructed to sit still and to remain silent. During the EEG study, children performed sentence-picture matching tasks. The picture-sentence matching task was presented by means of an original program, “Gramconstructor” (Certificate of state registration of computer programs No.2020616013, Saint Petersburg, Russia). On the laptop screen (17.3”), paired plot images were displayed in front of the children. A sentence was then fed into the headphones. At the end of the sound of a sentence, a question mark appeared on the screen, after which the subject selected one of the two pictures corresponding to the heard sentence and pressed the button on the keyboard below the selected picture. The duration of the test varied, since the transition to the next slide occurred in self-paced mode, i.e., there was no time limit for the answer. Sample structure: paired story images 0 to 2000 ms, first word in a sentence (NP1) 2000 to 3000 ms, second word (V) 3000 to 4100 ms, third word (NP2) 4100 to 5100 ms, question sign for motor response started from 5100 ms (Figure 1). For the analysis, the sentence had a marking of the beginning of each word in the phrase, with intonation and pauses characteristic of natural speech.

The whole experimental session was divided into four blocks, each containing 64 sentences in a pseudorandomized order. The pictures were randomly distributed among the blocks, but each block contained the same number of active and passive voice sentences). Each block lasted about 11.5 min. A break was included between blocks and additional breaks, if necessary. In total, one complete experimental session did not exceed 40 min.

### 2.5. Data Processing and Analysis

#### 2.5.1. Behavioral Data

The accuracy of thematic roles assignment in PV sentences was assessed by the percentage of correct answers; results ranged from 40 to 98.5%. Fifty of the 71 four to six-year-old children showed a level of accuracy of more than 60% of correct answers (Table 1). The final sample of the behavioral part of the experiment consisted of 14 four-year-old’s (age range 50–58 months, M = 55.5 months, six girls), 17 five-year-old’s (age range 60–70 months, M = 64.4 months, eight girls), and 19 six-year-olds (age range 71–83 months, M = 77 months, nine girls).

The data of one four-year-old (out of 14), three five-year-old’s (out of 17), and three six-year-olds (out of 19) in the EEG experiment were excluded because they contained disproportionate numbers of movement artifacts. The final sample of the ERP experiment consisted of 13 four-year-old’s, 14 five-year-old’s, and 16 six-year-olds. ERP results of six out of 13 four-year-old’s. and 11 out of 14 five-year-old’s were averaged in a previous study in the mixed four to five year old group of children with high (>80%) comprehension level of thematic roles assignment [62].

A statistical analysis was run in IBM SPSS Statistics version 26. A Shapiro-Wilk test was used to assess the normality of data distribution. As not all studied parameters showed normal distributions, significant differences in parameters were identified in the three age groups using the nonparametric Kruskal-Wallis H-test. Post-hoc analyses were run using the Mann-Whitney U-test with the Bonferroni correction for the number of comparisons (three age groups). The text contains mean values and 95% confidence intervals of the mean.

In each age group, the percentage of correct answers and reaction time (RT) of the sentence-picture matching task was calculated and post-hoc analysis was made using the Wilcoxon matched-pairs test for each sentence type (AV vs. PV). Differences were considered statistically significant at *p* ≤ 0.05.

#### 2.5.2. EEG Data

EEG processing and calculation of event-related potentials (ERPs) were performed using the program “WinEEG” (version 2.140.113). To remove very slow drifts and muscle artifacts from the EEG, a digital band-pass filter ranging from 0.3 to 20 Hz (3 dB cutoff frequencies of 0.37 and 19.91 Hz) was applied. The independent component method was used to remove oculomotor artifacts and myograms [68]. Record fragments containing other types of recording artifacts were removed from processing based on visual analysis. Evoked potentials were calculated from the beginning of audio stimulus presentation for each of 31 leads (to construct topograms of evoked response peak amplitude distribution): for the first word of the sentence (0–1000 ms), for the second word—verb or participle (0–1100 ms), for the third word—(0–1000 ms).

The record fragments relevant to test trials were separately averaged for each condition (active and passive voice direct word order sentences) and each subject. The resulting mean number of averaged EEG fragments across subjects was 53 ± 10 artifact-free trials for each type of sentence.

The ERPs were averaged from the beginning of each word in the sentence: (NP1, V, NP2). Groups of EEG sites were averaged according to the ROIs most frequently identified in the description of speech perception. The ROI’s selection in our study is based on the most frequent mention of brain areas in the context of sentence processing: the frontal and temporoparietooccipital areas on the left, as well as the temporal cortex area on the right [10,18,36,37,38]. The frontocentral zone averaged—nine sites (*F*3, *F*4, *Fz*, *FC*3, *FC*4, *FC*z, *C*3, *C*4, *C*z), the temporoparietal zones averaged five sites in the right (*T*8, *TP*8, *P*8, *CP*4, *P*4) and five sites in the left hemisphere (*T*7, *TP*7, *P*7, *CP*3, *P*3). For each age group, we compared the mean amplitudes of the ERP components for each word in the sentence at consecutive 100 ms TW’s. 

According to the Shapiro-Wilk test, the normality of distribution was assessed. Because of the non-normal distribution of some parameters, the significant between-group differences in averaged amplitude were identified using a nonparametric Kruskal-Wallis H-test. Post-hoc analyses were run using the Mann-Whitney U-test and the Bonferroni correction for the number of comparisons (three age groups). In each age group, the average amplitudes AV/PV in each TW were compared using the Wilcoxon matched-pairs test. Differences were considered statistically significant at *p* ≤ 0.05.

## 3. Results

### 3.1. Behavioral Results

Phonological awareness and lexical competence in speech estimation, as well as non-verbal intelligence and verbal memory showed medium and high age results in all age groups, o, TD children were selected for the ERP study. Neither non-verbal intelligence nor working memory and attention factors correlated with success in identifying thematic roles. The results can be found in Appendix A. 

Analysis of the correct picture identification in the sentence-picture matching task over the total sample of four-to-six-year-old children (*n* = 71), revealed that 50 children had correct answers that exceeded 60% in PV sentences and 57 children had correct answers above 60% in AV sentences (Table 1). The percentage of children with correct answers to PV increased with age from 61% in at four years old to 83% at six years old. The percentage of children with correct answers in AV increased with age from 69.5% at four years old to 91% at six years old. For further analysis, we took the results of four-to-six-year-old children who showed above 60% accuracy in correct answers to PV sentences.

To assess the skill of thematic-role assignment in AV and PV sentences, a between conditions comparison was made. Children of all the groups performed AV sentences better than PV (Figure 2a). In each age group, significant differences were found in the percentage of correct answers between AV and PV sentences (Wilcoxon matched-pairs test: 4 y.o. Z = −1.85, *p* = 0.03, 5 y.o. Z = −3.62, *p* = 0.000, 6 y.o. Z = −2.628, *p* = 0.003). In children aged five and six, significant differences were also found in RT (5 y.o. Z = −2.391, *p* = 0.01, 6 y.o. Z = −3.018, *p* = 0.001), AV sentences were processed faster (Figure 2b).

From a developmental point of view, accuracy differed significantly between age groups: the independent ANOVA (grouping) variable by Kruskal-Wallis revealed differences in correct answers to AV (H = 14.373, *p* = 0.001) and PV (H = 12.714, *p* = 0.002). The RT differences were revealed only in AV (H = 7.113, *p* = 0.03). For AV sentences, significant differences were found between children aged six and four, six and five, four and five in the percentage of correct answers; in RT, significant differences were found between children aged six and four, and six and five, revealed by the Mann-Whitney U-test (Figure 2, Table 1). For PV sentences, children six-years-old significantly differed from children of four and five years old in correctness; six-year-olds differed from four-year-old’s in reaction time (Table 1).

### 3.2. ERP Results

ERPs calculated for each word in the sentence were averaged over the three ROIs (frontocentral zone—nine sites, temporoparietal zones averaged five sites in the right, and five sites in the left hemisphere) in 100-ms time windows (TW). 

In accordance with the hypothesis of our study, we expected to reveal significant differences in ERP for the presentation of second (verb/participle) and third (noun) words in a sentence, since it is their inflections that encode AV and PV. However, AV vs. PV comparison revealed significant differences only on NP2 (the third word, for example, AV: “Vnuk (Nom) obn’jal deda (Acc)” (The grandson hugged the grandfather), vs. “Vnuk (Nom) obn’jat dedom (Inst)” (The grandson is hugged by the grandfather) in each age group. 

In four-year-old children (*n* = 13) who completed the thematic-role assignment with more than 60% correct answers, significant differences in the ERP amplitude in AV vs. PV were revealed by the Wilcoxon matched-pairs test: in the TW 300–400 ms (in frontocentral ROI *Z* = −1.992, *p* = 0.02 and right temporoparietal ROI *Z* = −2.551, *p* = 0.004). In the TW of 400 to 600 ms—in the left temporoparietal ROI (*Z* = −2.271, *p* = 0.01) (Figure 3a). In five-year-old children (n = 14) significant differences in the ERP amplitude in AV vs. PV were revealed by the Wilcoxon matched pairs test: in the TW 200–400 ms (in frontocentral ROI *Z* = −3.296, *p* = 0.001 and left temporoparietal ROI *Z* = −2.417, *p* = 0.01, right temporoparietal ROI *Z* = −3.107, *p* = 0.001). In the TW of 500 to 600 ms—in the frontocentral ROI (*Z* = −1.664, *p* = 0.05) (Figure 3b). ERP amplitude was higher for PV sentences in the frontocentral and left temporoparietal regions in four and five year old’s (Figure 3a,b). In six-year-old children (*n* = 16), significant differences in the ERP amplitude in AV vs. PV were revealed in TW 200–300 ms (frontocentral ROI *Z* = −2.275, *p* = 0.01; left temporoparietal ROI: *Z* = −2.275, *p* = 0.01; right temporoparietal ROI: Z = −2.172, *p* = 0.01), also in TW 500–700 ms in right temporoparietal ROI (Z = −1.862, *p* = 0.03) (Figure 3c). This P600-like component was registered in only six-year-old children.

The amplitude in all TWs was higher for PV sentences, which is illustrated by both the graphs and the topograms (Figure 3).

It should be noted that the differences in ERP for active and passive voice are manifested in the group of four-year-old children starting from 300 ms, while in the group of five- and six-year-old children it started from 200 ms in the frontocentral zone. At the same time, in four-year-old children, sensitivity to passive voice is first recorded in the frontocentral region starting from 300 ms and then spreads to the temporoparietal zone on the left in the TW of 400–600 ms. From the age of five, this sensitivity has a generalized character, simultaneously capturing the frontocentral and temporoparietal regions of both hemispheres, which is illustrated both by graphs and topograms in Figure 3.

### 3.3. Age Differences

The Age factor influence the first (NP1), second (V), and third (NP2) words of the sentence were shown by Kruskal-Wallis analysis. It is important to say that the first- and second-word differences were revealed only in age analysis and showed no effect in intragroup AV/PV comparison. Differences in frontocentral ROI during the third word (NP2) AV processing in TW 700–900 ms (H = 6.09, *p* = 0.04) were revealed by the Kruskal-Wallis test. ERP significant differences were obtained between children aged six and four (TW 800–900 ms, U = 36, *p* = 0.005), and six and five (TW 700–800 ms, U = 51, *p* = 0.005). In the left temporoparietal ROI differences were revealed during the first AV word (NP1) processing in TW 300–400 ms (H = 7.05, *p* = 0.03) and PV in TW 600–700 ms (H = 6.57, *p* = 0.04). ERP significant differences were obtained between children aged six and five in AV sentence processing in TW 200–400 ms (U = 48, *p* = 0.01) and in AV sentence processing in TW 600–700 ms (U = 53, *p* = 0.01). In right temporoparietal ROI differences were revealed during the second word (V) processing PV in TW 200–300 ms (H = 6.25, *p* = 0.04). ERP significant differences were obtained between children aged four and five (U = 45, *p* = 0.01).

## 4. Discussion

The aim of this study was to describe the developmental changes of complex grammar, i.e., passive voice comprehension during thematic role assignment in preschool children. Both behavioral and ERP differences were revealed during the picture-sentence matching task in comparison of AV vs. PV sentences in children four to six years of age with above >60% level of accuracy in thematic-role assignment (Table 2, Figure 3).

### 4.1. Behavioral Results

It was shown that the ability to comprehend complex grammar including passive voice forms by age seven to eight years in Russian speaking children [18,27]. The syntax-based strategy during thematic-role assignment is established quite late in ontogeny, as its prevalence showed only in those nine and 10 years of age [16]. The passive voice mastering is still in progress in four-to-six-year-old children; some of them are successful in their comprehension, and some are still on the chance level. Here we demonstrated that passive voice comprehension skill improves with age: the percentage of children whose correct picture identification was above 60% in four-year-old’s was 61%, while in five-year-old’s it was 68%, and in six-year-olds it was 83% according to the results in PV sentences (Table 1). This is in concordance with studies that showed a competence increase in passive voice comprehension with age in children [27,34,69,70]. In our study the percentage of correct responses between PV and AV differed in all age groups, five- and six-year-olds also had between condition differences in RT. We can assume the transition to the new passive voice-processing mechanism most likely occurring between ages of five and six: there is a dramatic increase in the percentage of children who understand passive voice, from 68 to 83. The age factor demonstrated that six-year-olds differed from both four-year-old’s and five-year-old’s. The age of passive voice mastering varies in different languages [19,25,29,30,31]. Our previous data demonstrated that children five and six years of age are already quite successful in passive voice comprehension [71]; at the same time the acquisition process in Russian-speaking children is nonlinear, and even seven-to-eight-year-old TD children can have difficulties with passive voice or reverse word order, especially with syntactically complex, transitive sentences [27,34]. 

Usually, the oral speech comprehending in a child is guided by the context and semantic cues [17], whereas our experiment is designed to rule out reliance on other cues except for grammar. In this way, the design of our experiment differs from the thematic-role assignment in ordinary speech. That is, we analyze the automatization of the syntactic skills of passive voice comprehending since semantic-independent syntax activity is established at nine to ten years of age [16]. Perhaps it is the conditions of the test, i.e., the impossibility of relying on semantics, that explains why the percentage of correct answers in AV sentences, the understanding of which is not questioned, reaches 90% only in six-year-olds. Our results on active voice comprehension (Appendix A) are concordant with results [6,72] obtained in Spanish and German studies with a similar experimental design, so it reflected common developmental processes, independent of language. 

### 4.2. ERP Results

To our knowledge, this is the first ERP study in Russian on PV sentence processing in four-to-six-year-olds. We had assumed that AV vs. PV ERP differences would correlate with the grammatical changes that influenced semantics and appear on the second (V) and third (NP2) words in sentences with direct word order. AV and PV sentences differ in the verb suffix (obnja**l**/obnja**t**—“hugged”/“hugged by”), and the noun inflection—NP2 (vnuk**a**/vnuk**om**—“grandson”/“by grandson”). 

We did not obtain ERP differences on the second (V) word in the sentence. We also did not find the literature to be analogous to this effect or its absence. In the most developed series of studies in German, there is no way to find an analogue, since the verb in grammar constructions used in experiments remains unchanged. In English, the P600 and LAN components are shown in a situation of conflict in the agreement of verbs according to the number or irregular forms of the verb [73,74]. In our preliminary results concerning the second part of the experiment with reversed word order in active and passive sentences, we found an effect on the verb group, but in this study, we have not received enough results to give a convincing explanation for the lack of effect in the direct word order sentences.

Thus, the effect of the passive voice comprehension in direct word order sentences is reflected in the ERPs, demonstrating cerebral sensitivity to grammatical markers of voice, i.e., ERP amplitude differences on the NP2 presentation in AV and PV. Notably, the ERPs amplitude was higher for PV vs. AV (Figure 3). The higher ERP amplitude for PV could be related to the subjective complexity of the task and increased demands on brain resources [75]. Data obtained are confirmed by our results on adults and children four-to-five years old with different levels of passive voice comprehension [62], where we demonstrated ERP sensitivity of the frontocentral and temporoparietal cortical areas bilaterally to the grammatical markers of the voice. At the same time, the sensitivity of the left temporoparietal cortex was manifested only in those who were successful in passive voice comprehension—in adults and children with a high (over 80%) level of correct thematic-role identification [62]. The specific role of the left temporal cortex in syntax processing was shown for the anterior portion of the left superior temporal gyrus (left aSTG) as being involved in syntactic processing at the phrase level [41,42], and the left posterior superior temporal gyrus (pSTG) as being involved in sentence comprehension in three-to-six-year-olds [59]. 

In this study, brain sensitivity to grammar voice (i.e., differences in the ERP amplitude to AV and PV) was revealed in all ROIs with common and specific features in age groups (Figure 3). In the frontocentral regions, differences were found in TW 200–400 ms (Figure 3). This component is biphasic: first it is positive, then crossing the zero becomes negative. Similar biphasic early positivity (220–600 ms) and late negativity (750–1200 ms) time-locked to NP2 was described in German three-year-old’s during sentence comprehension [56]. 

Early frontal positivity differences peaking at about 300 ms were shown at the age of 48 months when comprehending normal/anomalous words, which supposedly reflected a general-purpose mechanism for processing the oddity, this component precedes the use of a later, more language-specific effect of LAN and P600 [57]. The oddity could play a role in the passive voice comprehension because of its lesser frequency in comparison to active [23]. 4.5-year-old children showed a frontocentral ERP positivity effect during NP2 processing indicating difficulties in syntactic integration [10]. In German children aged 48 months (four-years-old), early frontal positivity differences peaking at about 300 ms were shown during comprehension of normal/anomalous words [57]. The authors considered that the frontal positivity effect in children overlapped with the LAN-effect, which is expected in adults, as it could represent the same cognitive process as LAN but, given a different developmental stage of the cytoarchitecture of Broca’s area [58,59], was generated by different brain electrical activity. In adults, LAN was reported at both NP1 and NP2 in response to non-canonical unambiguous sentences [60].

The negativity following the positive component could be correlated with the negativity effect shown in German adults and children at the TW of 100–400 ms in the frontocentral regions during thematic-role assignment in subject/object-initial sentences. NP1 sensitivity to the object-subject role has a more negative value in object-initial sentences than subject-initial ones in adults and children from the age of six years [10]. In our experiment we also got more negative values to active (NP2- object) than passive (NP2- subject) sentences. 

We obtained between condition differences in late positivity in five-year-old children in frontocentral ROI and in six-year-old children in right temporoparietal ROI (Figure 3, Table 2). Later components such as P600 and LPC (later positive component) are associated with processes occurring after word perception, with the analyses of grammatical markers such as inflections [17,76], and with syntactic and semantic reanalyzes [17,77], semantic integration [78,79] or decision certainty [80]. This component sensitivity could be a sign of the more mature language-specific effect [57].

Brain sensitivity (AV vs. PV ERP differences) was revealed in all age groups of four-to-six-year-old children with above 60% level of accuracy in thematic roles assignment. At the same time, from a developmental point of view, our study has shown the age differences (both behavioral and ERP) for six-year-olds from both four-year-old’s and five-year-old’s. Age differences could be correlated to executive function maturation—attentional processes reflected in P300 (TW 200–300 and 300–400 ms). Together with behavioral data—the percentage of correct answers and RT differences, this data shows the special place for six-year-olds in passive voice comprehension. 

Thus, we have shown in four-to-six-year-old children the sensitivity to passive voice markers—noun inflections mostly in the left hemisphere. It is likely that LAN can be seen as a mechanism for distinguishing active and passive voice markers in the early stages of mastering syntactic analysis of reversible sentences. As children grow older, the contribution of LAN as a component decreases (see Figure 3 for children six-years-old, topograms), the left-dominance disappears, the component becomes more central, more like N400. Molinaro and colleagues, analyzing the differences between LAN and N400 from the literature data, concluded that LAN is elicited by transparent morpho-phonological cues to process syntactic mismatches, while N400 is elicited by elaborate lexical information [81]. N400 amplitude is inversely correlated with stimulus predictability [75]. It is likely that the mechanism we observed in children reflects the early stages of mastering the skill of passive voice comprehension with the gradual transformation of neural mechanisms with age, including brain maturation with intensive myelination, axon growth and increasing fibre density [59,82,83] that are associated with the developmental refinement of the dorsal syntax network, leading to increasing specialization of the frontal areas [84]. 

Functional connectivity data obtained in comparison of three- and six-year old’s show the specific role of LIFG in different parts during thematic-role assignment in this age period [59]. Case marking becomes more pivotal for sentence interpretation than word order from the age of five [6,8]. Only at the age of seven do children behave like adults by relying on both case marking and word order [7]. It is possible that at six years of age we observe the process of transition to the “adult-type” of the mechanism for determining thematic roles with a predominant reliance on syntax, but this process probably does not end even at seven to 10 years old, although this requires further research.

The differentiation of temporal and parietal areas in semantics and syntax analyses occurs with the increasing influence of frontal cortical areas in syntax analysis [40]. The improvement of grammar skills at this age is reflected in a gradual switch to P600 mechanisms in sentence analysis [62]. In this study, only six-year-olds begin to show differences between AV and PV on the P600 component. In contrast, adults performing similar tasks showed differences in P600 across all ROIs examined [62]. 

## 5. Conclusions

The goal of the present study was to trace in detail the age-related changes in the brain mechanisms that ensure the passive voice comprehension in children four-to-six-years-old. The percentage of children whose correct picture identification was above 60% in four-year-old’s was 61%, in five-year-old’s it was 68%, and in six-year-olds it was 83%. Cerebral sensitivity to grammar voice markers was shown in all children, and it was expressed in ERPs amplitude differences in the response to the third word (second noun—NP2) in PV vs. AV sentences. The biphasic positive-negative LAN-like component at 200–400 ms was widespread from the frontocentral to the temporoparietal ROIs. In four and five year old’s it was left sided, while in six year old’s it was central, more like N400. In six year old’s, P600 in the right temporoparietal area was also registered resembled the adult-like pattern of activation. The ERP amplitude was higher for a PV sentence, reflecting the increased brain involvement in the processing of a more complex construction. In other words, the functional maturity of frontocentral and the left temporoparietal structures and their connections is critical to sensitivity to grammatical markers of the voice and ensures successful thematic roles assignment. From the developmental point of view, our study has shown the age differences (behavioral and ERP) for six-year-olds from both four-year-old’s and five-year-old’s, reflecting the ongoing grammar acquisition and brain development as well as the possible transition to the “adult-like” syntax-based thematic role assignment. 

## Figures and Tables

**Figure 1 brainsci-12-00693-f001:**
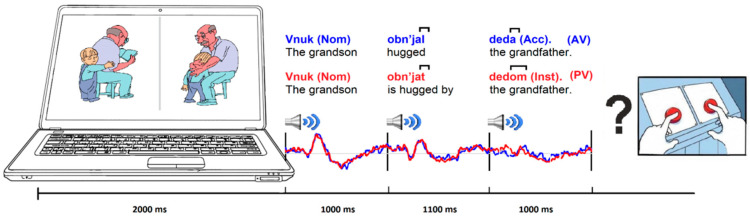
Experimental design in the sentence-picture matching task. Blue—active voice (AV), red—passive voice (PV), the frame marks morphemes that differed AV vs. PV sentences. Semantically distinctive morphemes are marked with square brackets. The vertical lines indicate the beginning of the stimulus sounding.

**Figure 2 brainsci-12-00693-f002:**
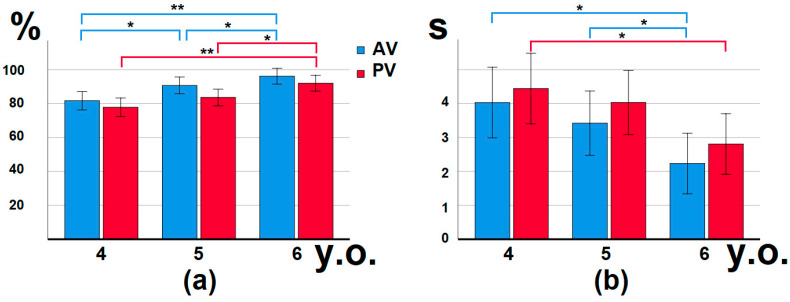
Correct identification (%) of a picture in the sentence-picture matching task (**a**) and RT (s) (**b**) to sentences with active (AV) and passive (PV) voice of four-to-six-year-old children, which had correct answers that exceeded 60% in PV sentences (*n* = 50), * *p* < 0.05, ** *p* < 0.001, 95% confidence interval.

**Figure 3 brainsci-12-00693-f003:**
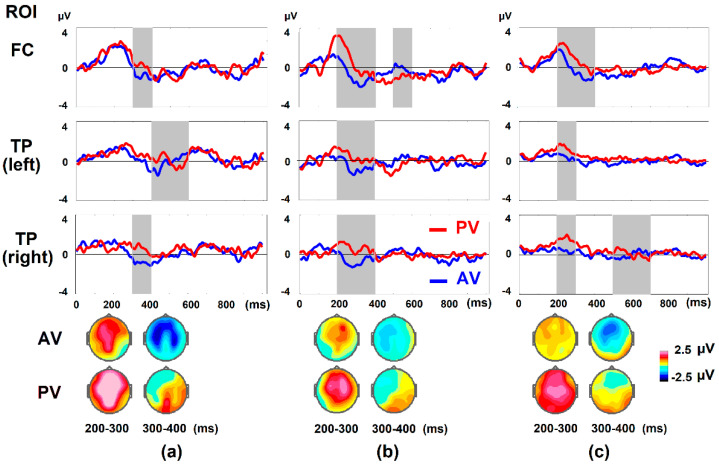
Average ERP amplitudes in the frontocentral (FC) and temporoparietal (TP left and TP right) ROIs from the beginning of NP2 in sentences with active (AV—blue) and passive (PV—red) voice. Time in ms is on the x-axis, and amplitude in µV is on the y-axis. Topography of ERP averaged for AV and PV in 100 ms TWs: 200–300 and 300–400 ms. (**a**)—four-year-old children (*n* = 13), (**b**)—five-year-old children (*n* = 14), (**c**)—six-year-old children (*n* = 16). The intervals of significant differences in the mean ERP amplitudes *p* < 0.05 are highlighted in gray.

**Table 1 brainsci-12-00693-t001:** Between-group behavioral differences were obtained in the sentence-picture matching task.

	4-year-old	5-year-old	6-year-old
The percentage of correct identifications			
4-year-old	*-*	PV: n.s.	PV: U *=* 47, *p* = 0.001
5-year-old	AV: U *=* 77, *p* = 0.05	-	PV: U *=* 77, *p* = 0.01
6-year-old	AV: U *=* 29.5, *p* = 0.000	AV: U *=* 98.5, *p* = 0.01	-
Reaction times			
4-year-old	*-*	PV: n.s.	PV: U *=* 78, *p* = 0.05
5-year-old	AV: n.s.	-	PV: n.s.
6-year-old	AV: U *=* 66, *p* = 0.01	AV: U *=* 98.5, *p* = 0.05	-

Notes. AV—active voice sentences, PV—passive voice sentences, U—the Mann-Whitney U-test, n.s. = non-significant.

**Table 2 brainsci-12-00693-t002:** Behavioral and ERP significant differences were obtained for the third word (NP2) of AV vs. PV sentences processing in age groups.

Age	Behavioral		ERP (NP2)		
	%	RT	FC	TP (Left)	TP (Right)
4 y.o.	+	−	TW 300–400 ms	TW 400–600 ms	TW 300–400 ms
5 y.o.	+	+	TW 200–400 ms TW 500–600 ms	TW 200–400 ms	TW 200–400 ms
6 y.o.	+	+	TW 200–300 ms	TW 200–300 ms	TW 200–300 ms TW 500–700 ms

Notes. %—the percentage of correct identifications, RT—reaction time, FC-frontocentral; TP (left)—left temporoparietal; TP (right)—right temporoparietal; TW—time window; significant (+) and non-significant (−) AV/PV differences according Wilcoxon matched-pairs test. For details behavioral results see Figure 2 and ERP results see Figure 3.

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
