# Peer review of "Passive Voice Comprehension during Thematic-Role Assignment in Russian-Speaking Children Aged 4–6 Is Reflected in the Sensitivity of ERP to Noun Inflections"

_brainsci, 2022, doi:10.3390/brainsci12060693_

Round 1
Reviewer 1 Report
lines 113-114. In addition to the number of children in the study sample, it would be useful to know the number of kindergartens that participated in the study and their characteristics. Considering the methodology that is the formative line of each school, it can be considered as a variable that influences the study.
Section 2.1. It is not known whether among the students in the sample there are any pupils with previous pathologies related to diction, phonological articulation or language problems.
Section 2.5 The reference to transmedia narratives is necessary, as it can contribute a great deal of significance to the meaning of the text. As an example, it may be interesting to include articles such as https://www.ceeol.com/search/article-detail?id=939300.
Section 4. It is recommended to differentiate between grammatical and phonetic results, which can benefit the understanding of the general assessment of the text.
Reviewer 2 Report
Dear authors,
The developmental trajectory of passive voice comprehension, particularly among multilingual children or between languages, is of interest to researchers in neurolinguistics. The approach used, attempting to develop a procedure that can assess comprehension without bias from other potential cues. Using ERP to evaluate potential neural correlates, and comparing children according to age and by task (passive vs. active voice sentences) is a meaningful approach. Much of the manuscript is written with clarity and detail, making replicability of the experiment possible. However, there are some considerations that I would like to point out which I believe are necessary to improve the quality and contribution of your manuscript. I will list these items below (first according to major points and then by section).
Best wishes in your current and future research.
Regards,
Reviewer
- Major points:
- The writing is unclear in many places. For example, abbreviations are introduced without providing the full form (including TD, NP1, V, NP2, ERP, RT, and ROIs in the abstract). Providing the full form of these abbreviations may be necessary to make the article’s content accessible to a wider range of readers. Moreover, for linguistic terms, there are some which could be more clearly described, including “cue acquisition” (line 38) or “inflection paradigms” (line 60). Sign-posting is also advised to clarify the logic and flow of the article.
- The development and statement of a clear gap and research question are lacking. I believe there is a gap/research problem being addressed, but it is not stated clearly and supported with some citations or logical development. In the paragraph starting on line 99, there needs to be a more clear process of clarifying the aims of the study and the resulting research question. In its present form “which is affect ERP mostly – the age changes itself or the skill?” (line 110), the RQ is unanswerable and misleading. That is to say, the use of cross-sectional data can provide some inferences into developmental trajectories, but “changes” themselves cannot be directly assessed. Moreover, it seems impossible to statistically or empirically evaluate the differential effects of age and passive/active voice accuracy and reaction time. As such, the RQ remains unanswered.
- In terms of the research design, the children are monolingual. I raise this issue since the Special Issue is “The Cognitive Science of Multilingualism.” Your study may contribute to neurolinguistics, overall, but cannot evaluate language acquisition among multilinguals or between languages.
- The grouping of participants by age is problematic. A four year old could be 48 months of age or 59 months of age. As such, the grouping by “year” is not discrete. A table of demographics, including average age (in months, with SD) for each of the three groups is recommended.
- Some aspects of the procedures are unclear. For example, the definition or justification of a cut off of 60% (line 230), “movement artifacts” (line 236), and “trials” (lines 259-263).
- The representation of the results is lacking in clarity. Figure 1 illustrates an example with two pictures. However, it is unclear what the right side picture represents, since it could be either the grandfather or grandson who is initiating the action. Table 1 has alignment issues and could present the post-hoc results a bit more clearly, without redundancies. Figure 2 requires labelling of the y-axis in terms of the measurement of reaction time (which unit?). Figure 3 contains a great deal of information, but would benefit from being displayed in a larger size. Also, the left side topography could be labelled more clearly. The illustrations of the topographical distribution of ERP difference activity could be larger, although I am not sure how you would draw conclusions from averaged distribution data. Table 2 is unnecessary, since this could be explained more succinctly using text. Table 3 is missing sufficient explanation of the +/- signs for the behavioral differences. It is also unclear how differences can be evaluated from this information, unless data from the Wilcoxon matched-pair tests is used as a supplement.
- There is no clear contribution beyond what is already known. In the discussion, many of the findings are mostly, such as the fact that the NP2 inflection (given its pivotal role) was the only element to register significantly based on ERP data, or that older children outperformed younger children in terms of PV comprehension (accuracy and RT). Other studies cited in this section suggest the reported results have already been fairly widely acknowledged (lines 411-415, 427-439, 446-452).
- In other cases, there needs to be more interpretation of specific findings, such as information on the ERP for specific ROIs or time windows, for example (lines 439-445, 453-460, 477-479, 480-491, 496-498).
- Avoid speculation, where your data cannot support the claims (lines 460-464, 501-509)
- Overall, based on the Discussion, I cannot see clearly how your RQ was answered. This means it is likely that you need to reformulate an RQ, based on a clear and definable gap in the literature. The content of lines 416-426 show some promise in formulating a more clear gap and RQ.
- Details by section:
Title:
- The title is too specific (thematic-role assignment based on syntax) and could be better served by including something more accessible, such as “passive voice comprehension.” Moreover, while the thematic-role assignment and the use of syntactic cues is the logical reason behind the findings, the findings themselves are more specifically and directly related to PV comprehension (accuracy and RT) only.
- The language (Russian) should be included.
- “Neural correlates” is mentioned, but a more specific emphasis was placed on neural functioning, which might be considered in terms of the title.
- Consider wording the title to reflect your key findings.
Abstract
- While a background is provided, there is no clear research problem, gap, or question.
- As mentioned above, too many abbreviations are included without being spelling out in full
- the use of “in” (line 13) suggests within-group differences rather than between/among-group differences.
- “voice grammar markers” (line 14) is out of context and may be unclear to the casual reader.
Keywords:
- “ERP” may need to be written in full
- “NP2” is not likely to be searched and, as such, is uninformative
- “frontocentral positivity” is only one of your findings related to the ROIs, what about others?
- Introduction
- You need to be careful to paraphrase content that was used from other sources (lines 39-41, 63-64, 78-80). As they stand, they are too similar to the original
- The nesting of parentheses (lines 50-52) is hard to follow. Reporting using standard linguistic formats is recommended, depending on journal formatting requirements
- Examples of “grammatical tools” used (line 56) would be of interest to readers
- The phrase “the age of its acquisition” (lines 58-59) is not clear. I do not understand your intended meaning
- If possible, cite researchers who note the need for more research on your target age group (lines 68-72). I do support this claim, but some substantiation would be helpful.
- Particularly if you are submitting to the Special Issue, more elaboration on different languages (line 71), including German (lines 73-82) would be expected. However, is it possible to make direct comparisons? For the particular ERP effects (positivity, biphasic components), more clear elaboration is needed at this point in the article.
- What is the language being evaluated in the content on lines 83-89?
- Caution is advised in making inferences about “developmental trajectory” (line 99), based on the cross-sectional nature of your date. The content that follows (lines 102-106) is helpful, but requires better elaboration.
- The research question (line 110), as mentioned above, is inappropriate. It cannot be evaluated, was not used to frame the Discussion, and omits several other factors which the data collection and analysis have emphasized.
- Research question, Materials and Methods
- A table of demographic data would be helpful. Moreover, since other data was collected, including gender (line 113), nonverbal intelligence (lines 140), and working memory and attention (line 141), could these variables be explored as potential factors?
- The development process of the “picture-sentence matching task” is not explained in enough detail. How were these develop? By whom? What are their qualifications? How was validity and reliability for these items established?
- The “counterbalance” of content according to certain “parameters” (lines 148-151) needs to be described in detail. What are these parameters?
- Validation for the stimuli is mentioned (lines 162 and 174), but the procedures are not clearly described.
- There is an error in the text (line 158). Please check the word order.
- If the design is to be balanced, as well as free from bias, can the decision to use a female speaker (line 177) be justified? Was there a specific reason for doing so?
- The nature of the “pseudorandomized order” (line 213) needs to be stated clearly.
- Take care to properly format sub-headings (Sections 2.2 to 2.6 could be under one subheading related to “materials,” while 2.10 should be under the “Data processing and analysis section.” There is no section for “EEG data”
- The cut off of 60% (line 230) should be cited or justified.
- “post-hoc” is misspelled (lines 241, 246, etc)
- The definition and examples of what is meant by “artifacts” (lines 253) and “trials” (lines 259-263) could be provided. This would help in more logically and clearly describing the inclusion and exclusion of participants and their data.
- The reason for the need for 33 artifact-free trials is not provided.
- The selection of the thee ROIs (lines 264-273) is reasonable, but could be briefly described in the Introduction and the most relevant neural correlates
- Results and discussion
- It is unclear what is meant by “we are going to do with a bigger sample” (line 286).
- In Figure 2, rather than using (a) and (b), a brief descriptor would be more reader-friendly. Also, as mentioned above, the units for the y-axis of the right side figure is not clear.
- Starting with the reporting of the importance of NP2 (line 323), it should be noted that this is not a surprising result, due to the nature of the language and the construction of the three word sentence, as well as the timing of the recorded audio.
- Discussion
- The assumption that the verb in the sentence would be related to ERP differences (line 379) was not mentioned earlier in the manuscript. This could be a fair assumption and, since this was not supported by the results, could be a potential area of contribution. For example, as compared to other languages, why does the inflected verb fail to result in ERP differences, as expected. If this could be further analyzed and interpreted, you might have a stronger research question and finding (even if that finding is not what was assumed).
- There is mention of participants who have a high (80%) level of PV comprehension (lines 387-390). This is worded as a finding from the study, but no data on this can be found in the manuscript. If this result is from another study, please include a citation.
- The organization of the section should be more logical, potentially with the use of subheadings, in order to more clearly identify findings by age and PV/AV comprehension (DVs), as well as ROIs, TWs, and positivity/negativity/biphasic aspects.
- The phrase “improves with age” may need to be changed as your study was cross-sectional. A safer statement might indicated that PV comprehension is associated with age, wherein older children have higher levels of comprehension.
- The design of your experiment, if you can (upon reflection) justify and explain how it is a contribution above and beyond previous studies, could be a contribution to be highlighted (lines 416-421)
- More interpretation of the amplitude and direction of ERP results (lines 434-445) should be provided, with a clear explanation of what positivity, negativity, and a biphasic component are indicative of, based on prior studies. That is, an interpretation of these findings (rather than simply confirming that they align with previous studies) would be of interest to readers.
- “4.5-year old” (line 450) seems to be a typo
- The results of citation [27] on lines 453-458 could be interpreted in more detail.
- Conclusions
- Again, interpretation of the biphasic component, from a neurolinguistic perspective, is lacking (lines 496-497).
- Please support the claim that the PV constructions used in your study are statistically “less frequent.” This is certainly the case, but a citation would help. This should also be noted earlier in the manuscript (introduction)
Author Response
Dear reviewer! Thank you very much for your careful reading and friendly attitude to our work, it is very important to us. Your questions and comments forced us to analyze our results and research questions again. We tried to answer all your questions as accurately as possible and change the text in accordance with your comments. Besides we consulted the native speaker (American) to correct our English. We hope this improves the quality of the study.

Round 2
Reviewer 2 Report
Reviewer feedback: Thank you for diligent response and useful revisions to the manuscript. As most of the comments/points have been address (either by making revisions or by explaining your rationale), I mostly have a few remaining points that I hope can assist in making further improvements to the manuscript. However, one main point that should be carefully considered is how to distinguish the present study with your previous publication, contextualizing this piece of your research in the context of your overall project. This is an important issue, and I ask for your consideration. I will provide more detailed comments and suggestions below. At first, I will respond to points and responses from the first round of review that I believe might need a bit more work. I will not address each response, since most of them have satisfied my concerns, but I do appreciate your effort and the time you invested in providing clear feedback to each point! After discussing previous points, I will provide a few more comments specific to this round.
Best regards,
Reviewer
Point 14. - You need to be careful to paraphrase content that was used from other sources (lines 39- 41, 63-64, 78-80). As they stand, they are too similar to the original
Response 14. Thank you for pointing on the way to improve our manuscript! We formulated the Introduction more correctly (lines 28-87).
Reviewer feedback: Overall, analysis by software indicates that 13% of the content of the present manuscript (not including references) overlaps with that of your previous publication [Kruchinina, O.V., Stankova, E.P., Guillemard, D.M. et al. The Level of Passive Voice Comprehension in the 4–5 Years Old Russian Children Reflects in the ERP’s. J Evol Biochem Phys 58, 395–409 (2022). https://doi.org/10.1134/S0022093022020089].
a) It is unavoidable that some of the procedures and background information from your previous publication is recycled in this manuscript, so that is not the principle concern.
b) The main concern is that some of the findings seem to be presented in both articles. Care should be taken not to report the same data in more than one publication (for ethical issues).
c) A secondary concern is that the key focus of the two studies contain many similarities (passive voice comprehension, young children, EEG data collection, the emphasis on NP2, the use of identical stimuli and participants [it seems], evaluation of ERPs in the same ROIs, and findings related to time windows for participants of different ages).
d) As such, if you are to use some of the same data and procedures from your previous publication, you need to be extremely cautious to ensure that this manuscript offers a clearly unique insight. A brief reading of the previously published article reveals that many of the key findings are repeated, particularly in terms of the neurological correlates of PV comprehension.
e) If you are able to ensure a clear and unique contribution, I suggest that you create a section under “1.4. The current research” which contextualizes this study in the overall research (which you note in Point 21 below) is an ongoing longitudinal design. As such, the overall goals of your project (lines 168 to 206) could be framed in terms of what data has been collected and analyzed thus far, the unique contribution of the present study, and future data collection and analysis plans. I believe that this would be necessary in order to “ethically” present the scope of your research, including how the current study relates to your previous publication and ongoing research.
Point 21. - Caution is advised in making inferences about “developmental trajectory” (line 99), based on the cross-sectional nature of your date. The content that follows (lines 102-106) is helpful, but requires better elaboration.
Response 21. Thank you for this point. We agree it is important. Strictly speaking, only longitudinal studies can claim the results of developmental trajectories; however, in studies on children, this formula has been used quite often, regardless of the method by which the results are obtained. In most cases, this concerns morphological and morpho -functional studies. By using this term, we wanted to emphasize that it is the process of the formation of functions that is important to us. We are going to make our study longitudinal; this study is only the first step. Still not sure if we are right to use it, but we would like to keep it as other options seem less suitable.”
Reviewer feedback: Please refer to the above point and attempt to position and describe your submitted manuscript in the context of your previously published work.
Point 22. - The research question (line 110), as mentioned above, is inappropriate. It cannot be evaluated, was not used to frame the Discussion, and omits several other factors which the data collection and analysis have emphasized.
Response 22. Special thanks to this comment. It forced us to analyse and elaborate our study. We rewrite the introduction and discussion sections. And we formulated new RQ. We hope it made our study more clear.
Reviewer feedback: This is helpful and makes the potential contribution more clear. However, the terminology of “developmental trajectory” (lines 157 and 207) “longitudinal study” (line 168) are misleading, as the present study is longitudinal. Thus, while your ultimate goal (after more data collection) is to be able to provide stronger evidence based on longitudinal data, the specific gap addressed by this study must be worded more carefully. The research aim of the present study (line 207) should be clarified even more to, to more clearly delineate which questions can be answered by the cross-sectional data you collected. In sum, more careful wording and clear description of your RQ is needed.
Point 23.- A table of demographic data would be helpful. Moreover, since other data was collected, including gender (line 113), nonverbal intelligence (lines 140), and working memory and attention (line 141), could these variables be explored as potential factors?
Response 23. Thank you for this notice! We added the demographic data in months (line 735) and Table S1 (behavioral, psychological and speech development assessments result in 3 groups of children). We did not reveal the influence of gender or nonverbal intelligence or working memory and attention factors on the success of determining thematic roles. We collected data on speech development, nonverbal intelligence, working memory and attention in order to include only children with typical development in the study.
Reviewer feedback: This is reasonable. Can you include this information in the manuscript?
Point 28. - If the design is to be balanced, as well as free from bias, can the decision to use a female speaker (line 177) be justified? Was there a specific reason for doing so?
Response 28. Thank you for this notice! The reviewer is right about the dictor’s gender influence, but we cannot rule it out. At the same time, a change of the speaker during the study may also have an effect, especially on the evoked response, because it is a biologically significant signal - a change in the environment that triggers an orienting reflex. Changes in tone and pitch of voice are reflected in the ERP characteristics. Therefore, we decided not to change the speaker and chose the female voice as the one most frequently heard by younger children. For review see https://doi.org/10.3389/fpsyg.2022.869475
Reviewer feedback: This is reasonable. Can you include this information in the manuscript?
Further points:
1. The content on Line 79 is a useful addition. However, in the manuscript (a) and (b) are not described. Also consider whether or not this could be inserted as a figure.
2. “ipm” (line 321) should be spelled out. I assume it is “instances per million”
3. The content on lines 394-396 is not clear. This needs to be reworded.
4. It appears some useful content was deleted and not translated: Как стимульный материал (объект действия, субъект действия, сюжетное 404 изображение, аудиофайл) так и процедура исследования были проверены на 405 отдельных группах детей. На одной группе детей (35 чел) проверили 406 предметные изображения
5. How was your study characterized as having “environmental friendliness” (lines 437-438). This is confusing.
6. The correct spelling of 2nd (lines 12, 570, 611, 613, 623, 694) and 3rd (lines 570, 611, 614, 635) should be use
Author Response
Many thanks to the reviewer for the attentive attitude to our study. We greatly appreciate the efforts of the reviewer to improve our manuscript. We hope that the study has become more clearly formulated and understandable as a result. We tried to take into account all the reviewer's comments, we edited the Introduction, and more clearly outlined the goals and questions of the study. We tried to distinguish between the previously obtained results and those presented here. We added clarifications in accordance with the reviewer's recommendations and corrected the English language in the Introduction and Discussion. More clearly rewritten Conclusion.
Best regards, authors.
